# Self-Evolving Critique Abilities in Large Language Models

Zhengyang Tang[1,4*]   Ziniu Li[1,2*]   Zhenyang Xiao[1*]   Tian Ding[2,3†]   Ruoyu Sun[1,2,3]
Benyou Wang[1†]   Dayiheng Liu[4†]   Fei Huang[4]   Tianyu Liu[4]   Bowen Yu[4]   Junyang Lin[4]

[1]The Chinese University of Hong Kong, Shenzhen
[2]Shenzhen Research Institute of Big Data
[3]Shenzhen International Center for Industrial and Applied Mathematics
[4]Qwen Team, Alibaba Inc.
dingtian@sribd.cn, wangbenyou@cuhk.edu.cn, liudayiheng.ldyh@alibaba-inc.com

## Abstract

Despite their remarkable performance, Large Language Models (LLMs) face a critical challenge: providing feedback for tasks where human evaluation is difficult or where LLMs potentially outperform humans. In such scenarios, leveraging the *critique* ability of LLMs themselves—identifying and correcting flaws—shows considerable promise. This paper explores enhancing critique abilities of LLMs, noting that current approaches rely on human annotations or more powerful models, leaving the challenge of improving critique abilities *without* external supervision *unresolved*. We introduce SCRIT (Self-evolving CRITic), a framework that trains LLMs with self-generated data to evolve their critique abilities. To address the low quality of naively generated data, we propose a contrastive-critic approach that uses reference solutions during data synthesis to enhance the model's understanding of key concepts, and incorporates a self-validation scheme to ensure data quality. The final trained model operates without any reference solutions at inference time. Implemented with Qwen2.5-72B-Instruct, a leading LLM, SCRIT demonstrates consistent improvements across a wide range of benchmarks spanning both mathematical and scientific reasoning: achieving a 10.0% relative gain in critique-correction accuracy and a 19.0% relative improvement in error identification F1-score. Our analysis reveals that SCRIT's performance scales positively with data and model size and enables continuous improvement through multi-round iterations.

## 1 Introduction

Large Language Models (LLMs) (Achiam et al., 2023; Anthropic, 2024; Qwen-Team, 2024) represent significant milestones in the development of artificial intelligence. They rely on human supervision through methods such as Supervised Fine-Tuning (SFT) (Wei et al., 2021; Li et al., 2025) and Reinforcement Learning from Human Feedback (RLHF) (Ouyang et al., 2022; Bai et al., 2022; Li et al., 2024). As a result, these models have evolved at an unprecedented pace, surpassing human capabilities in certain challenging domains. However, this framework encounters a fundamental challenge: how to provide effective and scalable feedback for LLMs in tasks that are not only difficult for humans to evaluate but where LLMs may outperform humans. This challenge, known as scalable oversight (Bowman et al., 2022), remains critical, yet progress in this area has been limited.

To address this challenge, leveraging LLMs for evaluation can help refine model outputs (Saunders et al., 2022; McAleese et al., 2024). Central to this approach is the *critique* ability—identifying and correcting flaws in responses. Accurate critique feedback enables LLMs to improve, advancing toward higher-order intelligence. Yet, studies show LLMs underperform in critique tasks (Zheng et al., 2024b; Yang et al., 2024; Tang et al., 2025). Thus, enhancing critique abilities is a key research problem, which this paper aims to tackle.

---

*Equal contribution.
†Corresponding author.

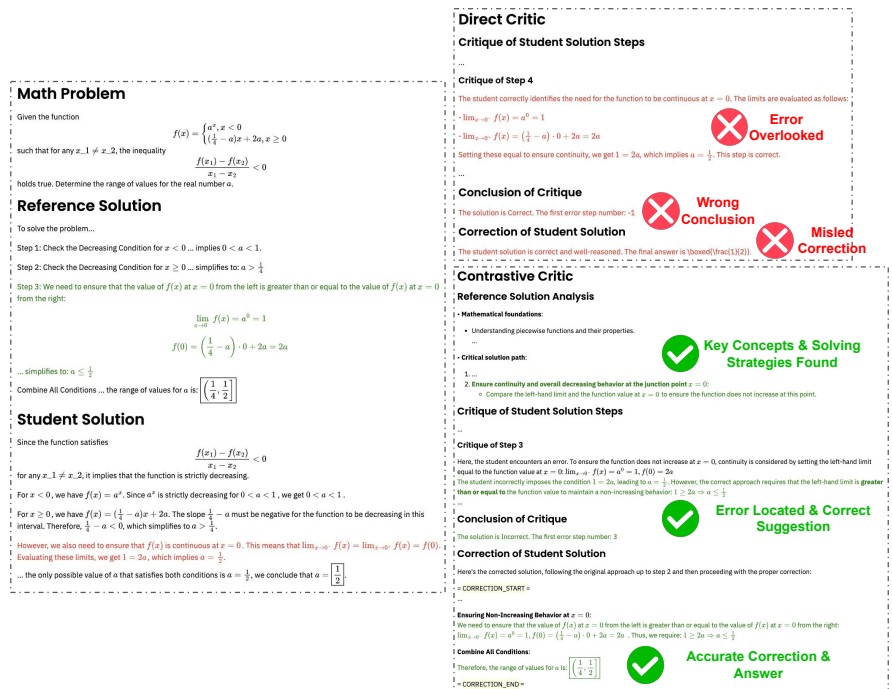

Figure 1: Direct critic (baseline) v.s. contrastive critic (ours). Left panel: input materials prepared for critique generation. Right panel: outputs from both approaches. The direct critic exhibits "rubber-stamping" behavior, incorrectly validating flawed solutions and providing misled feedback. The contrastive critic, however, utilizes reference solutions to grasp key concepts and strategies, enabling accurate error identification and correction.

Current approaches to improving the critique abilities of LLMs rely on two sources of supervision: human annotations (Saunders et al., 2022; McAleese et al., 2024) and stronger LLMs that serve as human proxy (e.g., GPT-4 and o1-mini) (Lan et al., 2024; Zhang et al., 2024; Zheng et al., 2024b; Yang et al., 2024)). While these methods have shown promise, they face three fundamental limitations. First, the quality of generated critiques is inherently bounded by the capabilities of the supervisors. Second, the dependence on human annotations or API calls to stronger models introduces significant costs, limiting the scalability of these approaches. Most critically, these approaches fail to address a fundamental question in scalable oversight: how can we enhance the critique abilities of our most capable models when stronger supervisors are no longer available?

In this work, we introduce SCRIT (Self-evolving CRITic), a framework that enables LLMs to develop self-evolving critique abilities in domains with verifiable solutions. We focus on **mathematical and scientific reasoning** as ideal testbeds for this approach. A key insight is that problems in these domains typically have well-defined reference solutions and corresponding final answers. These resources, leveraged *only during the data synthesis phase*, guide the critique of a student's solution and help verify the quality of the generated critique.

Our framework has two key steps to generate high-quality critique data for self-training.

- First, we develop a contrastive critique technique, where the model is provided with a reference solution to analyze and critique a student's solution. This step is grounded in our first philosophy: by conditioning on a correct reference solution first, the LLM develops a comprehensive understanding of the relevant concepts and problem-solving strategies, allowing it to accurately identify and address errors in student solutions. Our evidence shows that without this reference point, the model tends to exhibit "rubber-stamping" behavior—uncritically approving incorrect solutions and offering misleading feedback (see Figure 1 for examples).

- Next, the LLM is tasked with self-validating the generated critique to improve the data quality. Specifically, the model checks whether the proposed corrections lead to valid

solutions. This step is based on our second philosophy: critiques that result in internally consistent and correct correction are considered high-quality, which has also been widely adopted by recent works (Zheng et al., 2024b; Yang et al., 2024; Tang et al., 2025).

These two steps together enable the generation of high-quality critique data without human supervisions in writing good critiques for student solutions. Finally, we leverage the generated data to enhance the model's critique abilities through self-training. We clarify that while our framework requires reference solutions as input, it does not depend on ground truth critiques themselves, thus remaining within the scalable oversight paradigm.

We use Qwen2.5-72B-Instruct (Qwen-Team, 2024), a leading 70B model, to implement SCRIT. Our goal is to test whether our framework can further improve its performance. It is important to note that this is a non-trivial task, as Qwen2.5-72B-Instruct has already undergone extensive pre-training and post-training. Experiments show that SCRIT enables substantial improvements across different evaluation protocols as shown in Tables 1 and 2.

- On critic and correction tasks from (Tang et al., 2025), spanning 8 mathematical reasoning datasets across 3 scenarios, SCRIT consistently enhances the base Qwen2.5-72B-Instruct model: improving from 39.7% to 50.0% on deliberately incorrect solutions, from 57.7% to 62.1% on balanced solutions, and from 61.7% to 62.9% on the base model's self-generated solutions, representing a 10.0% relative gain in critique-correction accuracy on average.
- For error identification tasks on PRM800K (Lightman et al., 2023) and Process-Bench (Zheng et al., 2024a), two benchmarks with human-labeled error steps, SCRIT achieves consistent improvements across all datasets, raising the average F1 score from 37.8% to 45.0%, a 19.0% relative improvement.

In addition to these advancements, we provide a systematic analysis, which will be elaborated on in the main text. Our framework and methodology are detailed in the subsequent sections. Due to space constraints, the related work is discussed in Appendix A.

## 2 SCRIT: Self-Evolving Critic

### 2.1 Problem Formulation and Overview

Let $\mathcal{P}$ denote a set of problems from a structured domain (e.g., mathematics, science), where each problem $p \in \mathcal{P}$ is paired with an answer $a_p$. For each problem $p$, we collect a set of solutions $\mathcal{S}_p = \{s_1, s_2, ..., s_n\}$ from different models, where each solution $s_i$ consists of:

- A sequence of reasoning steps $\mathbf{r}_i = [r_i^1, r_i^2, ..., r_i^{k_i}]$, where $k_i$ is the number of steps
- A final answer $a_{s_i}$

A critique $c$ is defined as a tuple $c = (\mathbf{e}, l, t)$, where:

- $\mathbf{e} = [e_1, e_2, ..., e_k]$ is a sequence of step-wise critiques, where each $e_i$ corresponds to the analysis of step $r^i$
- $l = (y, j)$ is the conclusion, where $y \in \{0, 1\}$ indicates solution correctness and $j \in \{-1\} \cup \mathbb{N}$ denotes the first error step ($j = -1$ means no error)
- $t$ is the correction, consisting of a sequence of corrected steps and a final answer $a_t$

Our objective is to learn a critique function $f_\theta : \mathcal{P} \times \mathcal{S} \to \mathcal{C}$ that maps a problem $p$ and a solution $s$ to an effective critique $c$, where $\theta$ denotes the parameter to learn.

To achieve this objective, we propose SCRIT (Self-evolving CRITic), a framework that systematically leverages the shared mathematical understanding across different solutions to enable truly self-evolving critique abilities. SCRIT operates through a complete self-evolving cycle: it takes a problem and solutions as input, generates critiques through analyzing reference solutions, validates their quality, and uses the validated critiques for self-training. This forms a complete self-evolving cycle without any external supervision.

### 2.2 Solution Collection

**Dataset** The first step in our framework is to collect a diverse set of solutions. We build our collection process on the NuminaMath dataset (LI et al., 2024), a large-scale mathematical problem dataset covering various topics from elementary mathematics to competition-level

problems. To ensure data quality, we develop a robust pipeline to compute reliable ground truth answers (detailed in Appendix B), resulting in 452K validated problem-answer pairs.

**Solution Generation Models** To enhance the diversity of generated data, we gather solutions from seven models: deepseek-math-7b-rl (Shao et al., 2024), mathstral-7b-v0.1 (Mistral-AI, 2024a), Mistral-Large-Instruct-2411 (Mistral-AI, 2024b), DeepSeek-V2-Chat-0628 (DeepSeek-AI, 2024), Qwen2.5-Math-7B-Instruct (Qwen-Team, 2024), Qwen2.5-Math-1.5B-Instruct (Qwen-Team, 2024), and Qwen2-Math-1.5B-Instruct (Qwen-Team, 2024). It is important to note that the outputs from these models serve as inputs for the critic model, with no external supervision involved in the critic's learning process.

**Data Filtering** For each problem $p \in \mathcal{P}$, we classify its collected solutions into correct solutions $\mathcal{S}_p^+$ and incorrect solutions $\mathcal{S}_p^-$ based on answer correctness. A crucial filtering criterion in our framework is that each problem must have at least one correct solution and one incorrect solution to enable later contrastive critic. Formally, we only retain problems that satisfy: $\mathcal{P}_{valid} = \{p \in \mathcal{P} | |\mathcal{S}_p^+| > 0 \wedge |\mathcal{S}_p^-| > 0\}$.

## 2.3 Self-Critic Generation

A key challenge in enabling effective critique generation is to ensure the model can identify and correct errors in complex mathematical reasoning, particularly when the problem difficulty approaches or exceeds the model's current capabilities. Our preliminary experiments reveal that the model often exhibits "rubber-stamping behavior" - blindly approving incorrect steps without genuine understanding of the mathematical concepts involved, as illustrated in Figures 1 and 5. This also aligns with findings in (Huang et al., 2023).

We initially explored two approaches from previous works: (1) **Direct Critic** (Zheng et al., 2024a), where a language model directly critiques a solution; and (2) **Bug-Injection Critic** (McAleese et al., 2024), a two-stage approach of first injecting errors into a correct solution and then ask the LLM to critic and correct it. However, both approaches showed limited effectiveness (detailed in Section 4.3).

To address these issues, we develop a new technique called **Contrastive Critic**. Our key insight stems from a fundamental property of mathematical reasoning: *while problems may have multiple valid solutions, they share the same underlying mathematical concepts and key solving strategies*. By explicitly providing a correct reference solution during critique generation, we enable the model to first understand these core mathematical concepts and solving strategies, then leverage this understanding to perform step-by-step critique of the target solution. This approach addresses the rubber-stamping issue by grounding the critique process in concrete mathematical understanding derived from correct references.

For each valid problem, we generate critiques using two solution pairing approaches:

- **Correct-Incorrect Pairs.** For each incorrect solution, we randomly select a correct reference solution and generate a critique by comparing the incorrect solution against the reference.

- **Correct-Correct Pairs.** For each correct solution, we randomly select a different correct solution as reference and generate a critique comparing the two.

Both pairing strategies promote diversity in the generated critiques, which we empirically validate for effectiveness in subsequent experiments. The self-critic function (prompt template in Appendix C) decomposes critique generation into four sequential stages. **Stage 1 (Reference Analysis):** Generate a reference analysis that captures key mathematical concepts, critical solution steps, and potential pitfalls. **Stage 2 (Step-wise Critique):** For each step in the solution, generate a critique by verifying mathematical and logical validity using the reference analysis, identifying error type and suggesting corrections if found, and stopping analysis upon first error detection. **Stage 3 (Conclusion):** Generate a conclusion indicating both solution correctness (binary) and the first error step (if any). **Stage 4 (Correction):** Generate a correction by following the original approach up to the error step (if any), then completing with proper correction.

## 2.4 Self-Validation

With self-generated critique data, we apply post-validation techniques to further enhance the quality of generated outputs. This process specifically filters out low-quality cases where

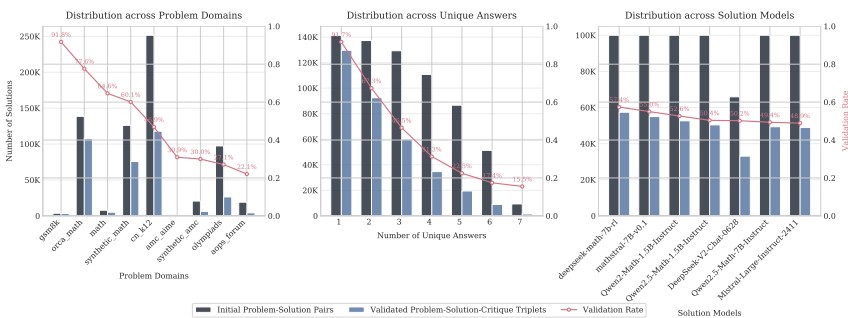

Figure 2: Data statistics before and after self-critic and self-validation filtering.

the model blindly approves all intermediate steps, only to suddenly reject the final answer upon detecting a discrepancy (see Appendix E).

To address these challenges, we employ direct validation on the correction part of the critique. Formally, we have that:

$$v_\theta(c) = \begin{cases} 1 & \text{if } g_\theta^l(p,t) = (1,-1) \\ 0 & \text{otherwise} \end{cases}$$

where $t$ is the correction part of critique $c$, and $g_\theta^l$ (prompt template in Appendix C) denotes direct critic's conclusion generation function that outputs a tuple $(y,j)$ as defined in Section 2.1. Here $g_\theta^l(p,t) = (1,-1)$ indicates that Direct Critic validates the correction $t$ as a fully correct solution with no errors ($y=1, j=-1$). This validation mechanism ensures that only critiques leading to verifiably correct solutions are used for self-training.

### 2.5 Self-Training

Let $\mathcal{V}$ denote the set of validated solution-critique pairs across all problems: $\mathcal{V} = \{(p,s,c) \mid p \in \mathcal{P}_{valid}, s \in \mathcal{S}_p, v_\theta(c) = 1\}$. For each validated triplet $(p,s,c)$, we construct a training instance. The input to the model consists of the problem $p$ and the student solution $s$. The target for fine-tuning consists of the critique components $(e,l,t)$ from $c$. Crucially, the reference solution used during data generation is **not** provided as input during training. This ensures the model learns a general critique ability independent of reference solutions at inference time. We then fine-tune Qwen2.5-72B-Instruct to minimize the cross-entropy loss.

## 3 Experiments

### 3.1 Statistics of SCRIT

We present detailed statistics of data flow through each component of our framework.

**Solution Collection** We start with 452K problem-answer pairs from our own NuminaMath dataset (see Appendix B). For solution generation, we employ 7 models of varying capabilities as described in Section 2.2. Each model generates one solution per problem, with solutions classified as correct or incorrect based on their final answers using Qwen2.5-72B-Instruct (detailed in Appendix I). Then we apply two filtering criteria: (a) Each problem must have at least one correct and one incorrect solution to enable contrastive learning; (b) Solutions from each model are capped at 50K for both correct and incorrect categories. After filtering, we obtain 665K problem-solution pairs, evenly split between good solutions (332K) and bad solutions (332K).

**Self-Critic & Self-Validation** To analyze the self-critic and self-validation step, we track the data flow from the initial 665K problem-solution pairs through these steps. Out of these pairs, 342K (51.4%) successfully pass the self-critic and self-validation step, yielding high-quality problem-solution-critique triplets. Detailed analysis in Figure 2 reveals systematic patterns: validation rates decrease from elementary domains (GSM8K: 91.8%, ORCA Math: 77.6%) to competition-level problems (Olympiads: 27.1%), show strong negative correlation with problem complexity (91.7% for single-answer problems to 15.5% for seven-answer problems), while remaining relatively consistent across solution models (48.9% to 57.4%).

Table 1: Performance comparison on Critic and Correct protocol.

| Model | RealCritic | | | | | | | | Avg. |
|---|---|---|---|---|---|---|---|---|---|
| | ARC-C | College Math | GPQA | GSM8K | MATH | Minerva Math | MMLU STEM | Olympiad Bench | |
| *Critic on deliberately incorrect solutions* | | | | | | | | | |
| Qwen2.5-72B-Instruct | 80.6 | 27.6 | 16.3 | 79.5 | 51.1 | 15.7 | 27.4 | 19.5 | 39.7 |
| + SCRIT | **86.7** | **32.6** | **25.3** | **88.3** | **66.0** | **23.4** | **50.7** | **27.0** | **50.0** |
| o1-mini | 74.9 | 34.8 | 26.3 | 88.6 | 78.0 | 23.8 | 45.5 | 40.8 | 51.6 |
| *Critic on balanced solutions* | | | | | | | | | |
| Qwen2.5-72B-Instruct | 85.2 | **50.9** | **31.1** | 88.3 | 72.0 | **47.1** | 42.1 | 44.6 | 57.7 |
| + SCRIT | **90.1** | 50.5 | 29.5 | **94.1** | **75.7** | 45.6 | **64.7** | **46.4** | **62.1** |
| o1-mini | 83.7 | 52.7 | 45.3 | 93.0 | 85.8 | 49.8 | 57.9 | 57.3 | 65.7 |
| *Critic on Qwen2.5-72B-Instruct's own solution* | | | | | | | | | |
| Qwen2.5-72B-Instruct | **93.5** | 45.9 | 32.6 | 96.7 | **83.6** | 38.3 | 59.6 | 43.4 | 61.7 |
| + SCRIT | 91.3 | 45.9 | **35.3** | 96.7 | 82.5 | **38.7** | **67.5** | **45.3** | **62.9** |
| o1-mini | 93.9 | 47.0 | 36.8 | 96.7 | 89.9 | 40.2 | 68.5 | 53.6 | 65.8 |

This suggests our self-validation process is more sensitive to problem difficulty than to the source model. Analysis of error positions in critiqued solutions (see Figure 16) reveals that a majority of errors occur in earlier steps, aligning well with human-labeled error distributions in ProcessBench (Zheng et al., 2024a). This correlation suggests that our self-critic framework captures human-like error identification patterns.

**Self-Training** We maintain a balanced 1:1 ratio between correct and incorrect solutions, resulting in 170K training examples. These balanced training data are used to fine-tune Qwen2.5-72B-Instruct following Section 2.5 (complete training details in Appendix J).

Table 2: Performance comparison on Critic and Correct with Error Identification protocol.

| Model | PRM800K | ProcessBench | | | | Avg. |
|---|---|---|---|---|---|---|
| | | GSM8K | MATH | Olympiad Bench | OmniMath | |
| Qwen2.5-72B-Instruct | 23.7 | 68.9 | 50.9 | 25.5 | 20.0 | 37.8 |
| + SCRIT | **24.6** | **80.2** | **60.0** | **32.5** | **27.8** | **45.0** |
| o1-mini | 34.0 | 88.0 | 81.1 | 53.0 | 38.6 | 58.9 |

## 3.2 Evaluation

We present two complementary evaluation protocols to assess critique capabilities:

**Critic and Correct** The first protocol evaluates a model's ability to critic and correct a given solution, following the assumption (Zheng et al., 2024b) that effective critiques should guide the correction of errors. We conduct experiments on RealCritic (Tang et al., 2025), which spans benchmarks from two key domains: **Mathematical Reasoning** (GSM8K (Cobbe et al., 2021), MATH (Hendrycks et al., 2021), College Math (Tang et al., 2024), Minerva Math (Lewkowycz et al., 2022), OlympiadBench (He et al., 2024)) and **Scientific Reasoning** (ARC-C (Clark et al., 2018), GPQA (Rein et al., 2023), MMLU-STEM (Hendrycks et al., 2020)). Evaluation is conducted across 3 scenarios: critic on incorrect solutions, balanced solutions, and the base model's self-generated solutions (i.e., Qwen2.5-72B-Instruct's own solutions).

**Critic and Correct with Error Identification** The second protocol requires models to provide accurate correction and identify the first error step. We evaluate on PRM800K (Lightman et al., 2023)[1] and ProcessBench (Zheng et al., 2024a), which contain human-labeled error steps from advanced models across GSM8K, MATH, OlympiadBench, and Omni-Math. Following ProcessBench, we use the F1 score of accuracies on incorrect and correct samples as our metric, with two adaptations to ensure critique effectiveness (See Appendix G).

**Baselines** Since our goal is to improve Qwen2.5-72B-Instruct's critique ability through self-evolution, we use the original Qwen2.5-72B-Instruct as our primary baseline. Additionally, we compare against o1-mini (OpenAI, 2024), currently one of the most capable models in terms of critique ability (Zheng et al., 2024a).

---

[1] https://github.com/openai/prm800k/blob/main/prm800k/data/phase2_test.jsonl

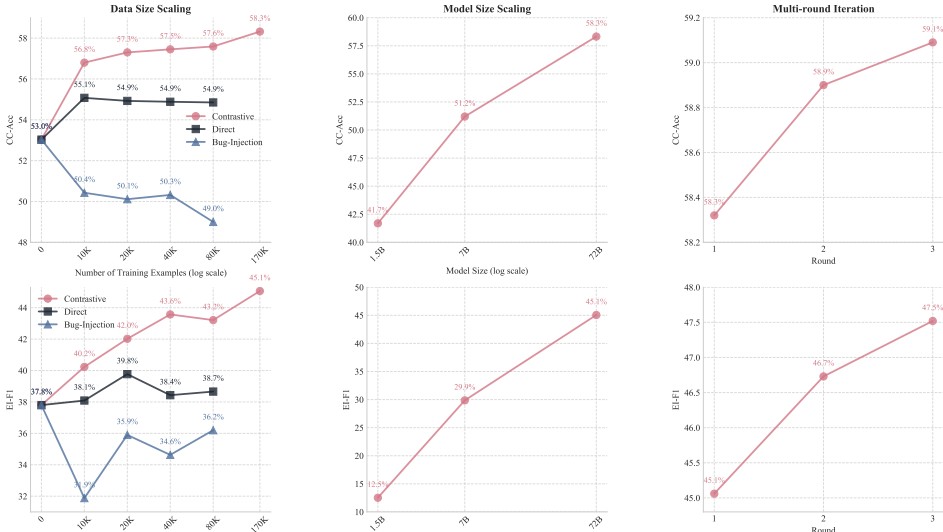

Figure 3: Scaling and multi-round performance analysis. **Left panel:** Data size scaling of Contrastive Critic, Direct Critic, and Bug-Injection Critic. **Middle panel:** Model size scaling from 1.5B to 72B parameters. **Right panel:** Multi-round self-evolving over 3 iterations.

### 3.3 Main Results

**Critic and Correct** Table 1 presents results across three increasingly challenging scenarios. SCRIT substantially improves over the base Qwen2.5-72B-Instruct model on deliberately incorrect solutions (50.0% vs 39.7%), maintains a 4.4% advantage on balanced solutions, and even improves when critiquing the base model's own solutions (62.9% vs 61.7%). These results represent a 10.0% relative gain in critique-correction accuracy across all scenarios, approaching the performance of o1-mini.

**Critic and Correct with Error Identification** As shown in Table 2, SCRIT shows strong capabilities in error identification, achieving consistent improvements across all datasets in both PRM800K and ProcessBench. The average F1 score improves from 37.8% to 45.0% (a 19.0% relative improvement), with strong gains on mathematical reasoning tasks (GSM8K: +11.3%, MATH: +9.1%). While there remains a gap with o1-mini, SCRIT's improvements are notable given its self-evolving nature without reliance on external supervision.

## 4 Analysis

Throughout this section, we report two metrics: critique-correction accuracy (CC-Acc) from the Critic and Correct protocol, which is averaged across three scenarios, and error identification F1-score (EI-F1) from the Critic and Correct with Error Identification protocol.

### 4.1 Generalization to Scientific Reasoning

A key question is whether SCRIT's self-evolving mechanism, primarily trained on mathematical data, can generalize to other complex reasoning domains. Our evaluation already includes scientific reasoning benchmarks (ARC-C, GPQA, MMLU-STEM), where Table 1 shows consistent improvements, confirming cross-domain generalization.

To further investigate this, we conducted an additional experiment where we trained a separate SCRIT model exclusively on 10K critique examples synthesized from scientific reasoning problems (from AM-Thinking-v1 (Ji et al., 2025)). As shown in Table 3, SCRIT trained on scientific data yields even stronger performance on scientific benchmarks (+10.6% on balanced solutions) while remaining competitive on math tasks. This not only demonstrates the framework's effectiveness beyond mathematics but also highlights its ability to capture and leverage domain-specific error patterns.

Table 3: Cross-domain generalization. SCRIT trained on domain-specific data shows strong in-domain performance and effective cross-domain transfer. CC-Acc is reported on balanced solutions.

| Model | CC-Acc (Balanced) | | Overall Avg |
| --- | --- | --- | --- |
| | Math Reasoning | Scientific Reasoning | |
| Qwen2.5-72B-Instruct | 60.2 | 52.8 | 57.7 |
| + SCRIT (Math Data) | **64.5** | 61.4 | 62.1 |
| + SCRIT (Scientific Data) | 61.8 | **67.4** | 63.9 |

### 4.2 Scaling Behavior of SCRIT

We investigate SCRIT's performance scaling with training data and model size (see Figure 3).

**Data Size Scaling** For data scaling experiments, we train SCRIT with different amounts of training examples, ranging from 10K to 170K. Both CC-Acc and EI-F1 show consistent improvements with increased training data. The CC-Acc improves from 53.0% to 58.3%, with the steepest gains in the early stage (0-20K examples) and continued but more gradual improvements afterwards. Similarly, EI-F1 increases from 37.8% to 45.1%, demonstrating that SCRIT can effectively leverage more training data to evolve its critique capabilities.

**Model Size Scaling** We evaluate SCRIT across three model sizes of Qwen2.5: 1.5B, 7B, and 72B. Both metrics show strong positive correlation with model scale. The CC-Acc increases substantially from 41.7% (1.5B) to 51.2% (7B) and further to 58.3% (72B). The improvement is more pronounced for EI-F1, where metric rises from 12.5% to 29.9% and then to 45.1%, suggesting that larger models are particularly better at error identification.

To further verify that SCRIT is not only beneficial for large models, we applied the framework to a mid-sized model, Qwen2.5-32B-Instruct. SCRIT improved its performance meaningfully, with CC-Acc increasing from 53.9% to 56.5% and EI-F1 from 35.8% to 41.5%. This shows that the self-evolving mechanism is robust and effective across different model scales.

### 4.3 Which Critic Mechanism is Most Effective?

To identify the most effective critic mechanism for our self-evolving framework, we conduct strictly controlled experiments comparing three different critic approaches described in Section 2.3 using identical sets of problems and solutions.

Our experiments in Figure 3 reveal several key findings. First, Contrastive Critic shows strong performance from the early stages across both metrics: with just 10K training examples, it achieves 56.8% CC-Acc and 40.2% EI-F1, outperforming both Direct Critic and Bug-Injection Critic. More importantly, as training data increases to 170K examples, Contrastive Critic continues to show positive scaling behavior, reaching 58.3% CC-Acc and 45.1% EI-F1. In contrast, Direct Critic quickly plateaus at around 55.1% CC-Acc and 38.7% EI-F1, while Bug-Injection Critic exhibits performance degradation in CC-Acc (dropping to 49.0%) and unstable performance in EI-F1).

Through case studies (detailed in Appendices D and F), we identify the key mechanisms behind these performance differences. Direct Critic often falls into superficial critiquing, tending to blindly agree with solutions without deep understanding. Contrastive Critic avoids this pitfall by first analyzing reference solutions, enabling the model to develop a deeper understanding of the underlying mathematical concepts and solution strategies before attempting critique. While Bug-Injection Critic has the theoretical advantage of known error descriptions, our analysis reveals that model-injected bugs tend to be simplistic and repetitive, predominantly focusing on basic arithmetic errors and variable confusions, limiting its effectiveness in real-world scenarios where errors are more diverse and subtle.

### 4.4 Does Multi-round Iteration Foster Improvement?

A unique advantage of SCRIT is its ability to support multi-round self-evolution. After collecting the initial solutions, we can iteratively apply the self-critic generation, self-validation, and self-training steps to continuously improve the model's critique abilities. Specifically,

Table 4: Controlled ablation studies on SCRIT. Each experiment varies only the target component while keeping all other settings fixed at baseline: 10K training examples with contrastive critic and self-validation, diverse domains, all solution models, and balanced solution ratio. Red/green numbers indicate the relative performance decrease/increase.

| Setting | CC-Acc | EI-F1 |
|---|---|---|
| Baseline | 56.8 | 40.2 |
| *Self-Validation* | | |
| Without Self-Validation | 56.0 (-0.8) | 37.2 (-3.0) |
| *Problem Domain* | | |
| Limited to GSM8K + MATH | 55.4 (-1.4) | 38.8 (-1.4) |
| *Problem Difficulty* | | |
| More Unique Answers First | 55.8 (-1.0) | 38.1 (-2.1) |
| Less Unique Answers First | 56.2 (-0.6) | 42.3 (+2.1) |
| *Single Solution Model* | | |
| deepseek-math-7b-rl | 56.5 (-0.3) | 39.8 (-0.4) |
| mathstral-7B-v0.1 | 56.0 (-0.8) | 39.2 (-1.0) |
| Mistral-Large-Instruct | 56.3 (-0.5) | 40.3 (+0.1) |
| DeepSeek-V2-Chat | 56.3 (-0.5) | 40.0 (-0.2) |
| Qwen2.5-Math-7B | 56.2 (-0.6) | 40.7 (+0.5) |
| Qwen2.5-Math-1.5B | 56.2 (-0.6) | 40.9 (+0.7) |
| Qwen2-Math-1.5B | 55.9 (-0.9) | 40.9 (+0.7) |
| *Good:Bad Solution Ratio* | | |
| 0.75:0.25 | 55.1 (-1.7) | 38.1 (-2.1) |
| 0.25:0.75 | 56.6 (-0.2) | 41.0 (+0.8) |

we conduct experiments with three rounds of iterations. Starting with Qwen2.5-72B-Instruct as the base model, we apply SCRIT to obtain an enhanced model with improved critique capabilities. As shown in Figure 3, using this enhanced model as the new base for Round 2, we observe further improvements in both metrics. Continuing with the Round 2 model for the third iteration, we achieve additional gains.

The performance demonstrates consistent positive scaling across both metrics through multiple rounds of iteration. This sustained improvement suggests that SCRIT can effectively leverage its own enhanced critique capabilities to generate increasingly higher-quality training data, enabling genuine self-evolution without external supervision.

## 4.5 How Important is Self-Validation?

To assess the necessity of self-validation in SCRIT, we conduct controlled experiments by removing the self-validation component while keeping all other settings identical. The results in Table 4 show clear performance degradation across both evaluation metrics: the CC-Acc drops by 0.8%, and more significantly, the EI-F1 decreases by 3.0%. Case analysis (see Appendix E) shows that the self-critic may still generate low-quality critiques, often blindly approving all intermediate steps only to suddenly claim "the final step is incorrect" when encountering answer discrepancies. By incorporating self-validation, we are able to further enhance the quality of data for self-training.

## 4.6 Does Problem Domain Diversity Matter?

To investigate the importance of problem domain diversity, we conduct controlled experiments by restricting the training data to only GSM8K and MATH, while keeping other settings unchanged. This represents a significant reduction in domain coverage compared to our full setting which spans 9 sources ranging from elementary to competition-level mathematics. The results in Table 4 show the value of domain diversity: when training with limited domains, the CC-Acc drops by 1.4% and the EI-F1 decreases by 1.4%. It suggests that exposure to diverse problem-solving patterns and error types is crucial for developing robust critique abilities.

### 4.7 How Does Problem Difficulty Impact Performance?

To understand the impact of problem difficulty, we conduct experiments by selecting training examples based on the number of unique answers generated across solution models - a proxy for problem complexity. We study two settings: training with problems that have more unique answers (indicating higher complexity) versus those with fewer unique answers (indicating lower complexity). Interestingly, training with less complex problems leads to better performance in EI-F1 in Table 4. This result suggests that SCRIT can generate more effective critiques on simpler problems, possibly because the mathematical concepts and solution strategies in these problems are more structured and well-defined, enabling the model to develop more precise and reliable critique patterns.

This finding leaves space for future work: how to optimally select training examples based on difficulty levels in a self-evolving framework. While our current approach uses all available data, a more sophisticated curriculum that gradually increases problem complexity might lead to more effective self-evolution.

### 4.8 Does the Choice of Solution Model Matter?

To study whether critiquing solutions from different models affects SCRIT's performance, we conduct controlled experiments by restricting the solutions being critiqued to those from a single model while keeping other settings identical. Our results in Table 4 show that the source model of solutions has limited impact on SCRIT's final performance.

Since solution generation models only provide the solutions for constructing contrastive critique pairs and do not directly participate in improving critique effectiveness, their individual capabilities have less influence on the final performance. What matters more is how to construct diverse and informative contrastive pairs that help the model learn effective critique strategies, regardless of the solution models.

### 4.9 Optimal Ratio between Good and Bad Solutions?

Finally, we investigate the impact of good-to-bad solution ratio in the training data. As shown in Table 4, training with a higher proportion of bad solutions (0.25:0.75) shows better performance than using more good solutions (0.75:0.25). This suggests that exposure to more bad solutions helps SCRIT develop stronger error identification capabilities, likely because it provides more diverse examples of mathematical mistakes and their corresponding corrections. More importantly, analyzing incorrect solutions forces the model to actively engage in error detection and correction, rather than simply validating correct steps.

## 5 Conclusion

In this work, we present SCRIT, a self-evolving critique framework that enhances critique-correction accuracy and error detection in domains with verifiable solutions. By leveraging a contrastive-critic mechanism during data synthesis, SCRIT improves its capabilities without external supervision. Our experiments, spanning both mathematical and scientific reasoning, demonstrate that SCRIT scales with data and model size, shows strong cross-domain generalization, and benefits from self-validation. Future work could consider using SCRIT's high-quality critiques to label reasoning steps and optimize student models via reinforcement learning (e.g., (Saunders et al., 2022; McAleese et al., 2024)), or extending the framework to other structured domains like coding and logic.

## Acknowledgments

The work of Tian Ding is supported by Hetao Shenzhen-Hong Kong Science and Technology Innovation Cooperation Zone Project (No.HZQSWS-KCCYB-2024016). The work of Ruoyu Sun is supported by NSFC (No. 12326608); Hetao Shenzhen-Hong Kong Science and Technology Innovation Cooperation Zone Project (No.HZQSWS-KCCYB-2024016); University Development Fund UDF01001491, the Chinese University of Hong Kong, Shenzhen; Guangdong Provincial Key Laboratory of Mathematical Foundations for Artificial Intelligence (2023B1212010001). The work of Benyou Wang is supported by Shenzhen Doctoral Startup Funding (RCBS20221008093330065), Tianyuan Fund for Mathematics of National Natural

Science Foundation of China (NSFC) (12326608), Shenzhen Science and Technology Program (Shenzhen Key Laboratory Grant No. ZDSYS20230626091302006), and Shenzhen Stability Science Program 2023, Shenzhen Key Lab of Multi-Modal Cognitive Computing.

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

## A  Related Work

**Scalable Oversight and Critic Models** The challenge of providing effective feedback to language models on tasks difficult for humans to evaluate has attracted significant research attention. Early work by (Saunders et al., 2022) proposed fine-tuning LLMs to generate natural language critiques, introducing key components including critique generation, discrimination, and correction. Building on this direction, CriticGPT (McAleese et al., 2024) applied similar principles to code review tasks, incorporating RLHF and specialized human supervision through a "Tampering" step. These works established the importance of critique ability in enabling scalable oversight of language models.

**Sources of Critique Supervision** Existing approaches to developing critique abilities primarily rely on two types of supervision sources. The first category uses human supervision, as demonstrated in (Saunders et al., 2022) through direct human annotation and in (McAleese et al., 2024) through human-injected errors. The second category employs strong model supervision, exemplified by MultiCritique (Lan et al., 2024), which utilizes feedback from advanced models like GPT-4 to generate critiques for fine-tuning smaller models. Recent work GenRM (Zhang et al., 2024) proposes Chain-of-Thought Verifiers that generate step-wise critiques for mathematical reasoning, though still relying on human or stronger model supervision. While these approaches have shown promise, they are fundamentally limited by either the capabilities of their supervisors or the substantial costs associated with obtaining supervision.

**Critic and Correct** An important challenge in developing critique systems is how to evaluate the quality of critiques themselves, as directly measuring critique effectiveness is often as difficult as the original task. A key insight that has emerged in recent work is that truly effective critiques should be able to guide the correction of errors and lead to correct answers. This assumption provides a validation mechanism for critique quality and has been widely adopted in the field. For instance, Critic-CoT (Zheng et al., 2024b) combines step-wise critique generation with correction validation using GPT4-Turbo. Similarly, SuperCorrect (Yang et al., 2024) collects critique and corrections from teacher models like o1-mini. These

works show the value of using correction as an objective mechanism to verify critique quality, though they still rely on stronger models for supervision.

In contrast to existing approaches that rely on either human annotations or stronger models for supervision, our work introduces SCRIT, a framework that enables self-evolution of critique abilities. By analyzing correct reference solutions to understand key mathematical concepts and strategies, then validating critiques through correction outcomes, our approach creates a closed-loop learning system that can improve its critique capabilities without external supervision.

## B Computing Ground Truth Answers for NuminaMath

A large-scale dataset with reliable ground truth answers is fundamental to our work. We choose NuminaMath (LI et al., 2024) for its diversity, difficulty distribution, and scale (860K problems). However, as the correctness of solutions in the original dataset is not guaranteed, we develop a robust pipeline to compute reliable ground truth answers.

### B.1 Answer Generation and Validation Pipeline

We employ Qwen2.5-Math-72B-Instruct (Qwen-Team, 2024) under tool-integrated (Gou et al., 2023) settings to generate solutions, as it demonstrates state-of-the-art performance across multiple mathematical reasoning benchmarks. The solutions are then evaluated using Qwen2.5-Math-RM-72B (Qwen-Team, 2024), a specialized reward model for mathematical reasoning. We consider a solution correct if its reward score exceeds a predefined threshold, and use its final answer as the ground truth.

### B.2 Threshold Selection and Validation

To determine an appropriate reward threshold, we conduct extensive experiments:

- **Benchmark Validation:** We evaluate the threshold's effectiveness across multiple standard benchmarks including GSM8K (Cobbe et al., 2021), MATH (Hendrycks et al., 2021), GAOKAO2023-EN (Zhang et al., 2023), OlympiadBench (He et al., 2024), and College Math (Tang et al., 2024). With a threshold of 1.0, we achieve approximately 75% accuracy.

- **Human Evaluation:** We randomly sample 100 NuminaMath problems and conduct human evaluation of the answers selected using our threshold. The results show approximately 85% accuracy.

- **Comparison with Alternative Methods:** We explore majority voting among solutions from NuminaMath, Qwen2.5-Math-72B-Instruct, and Deepseek-V2-Chat-0628. However, this approach yields lower accuracy compared to our reward-based selection method.

After applying our pipeline with the validated threshold, we obtain a filtered dataset of 452K problem-answer pairs, which serves as the foundation for our work.

## C Prompting Templates for Direct Critic, Bug-Injection Critic and Contrastive Critic

Here we present system prompts used for different critic mechanisms in Figure 4.

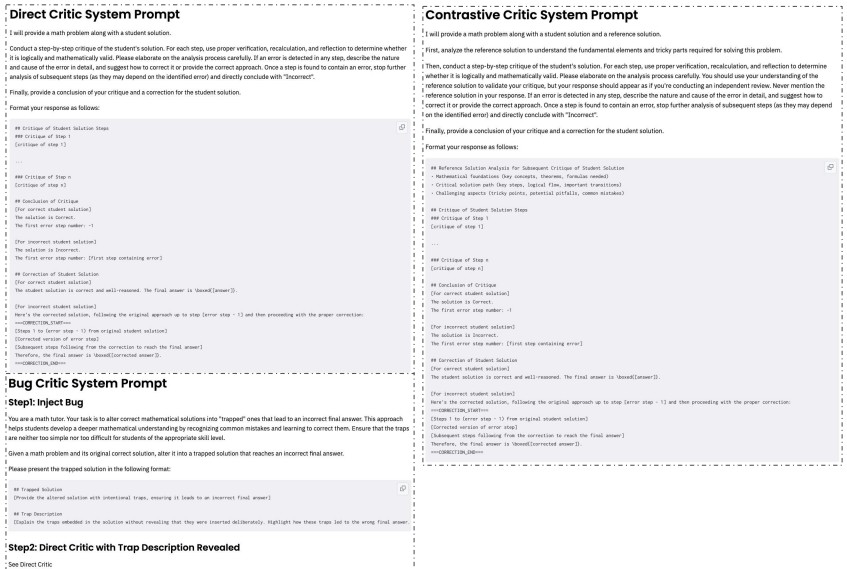

Figure 4: System prompts used for different critic mechanisms. **Top Left:** Direct Critic directly analyzes solution correctness without any additional context. **Bottom Left:** Bug-Injection Critic first injects bugs (Step 1) then direct critic on bug-injected solution (Step 2). **Right:** Contrastive Critic first analyzes a reference solution to understand key mathematical concepts before conducting step-wise critique.

## D    More Comparison between Direct Critic and Contrastive Critic

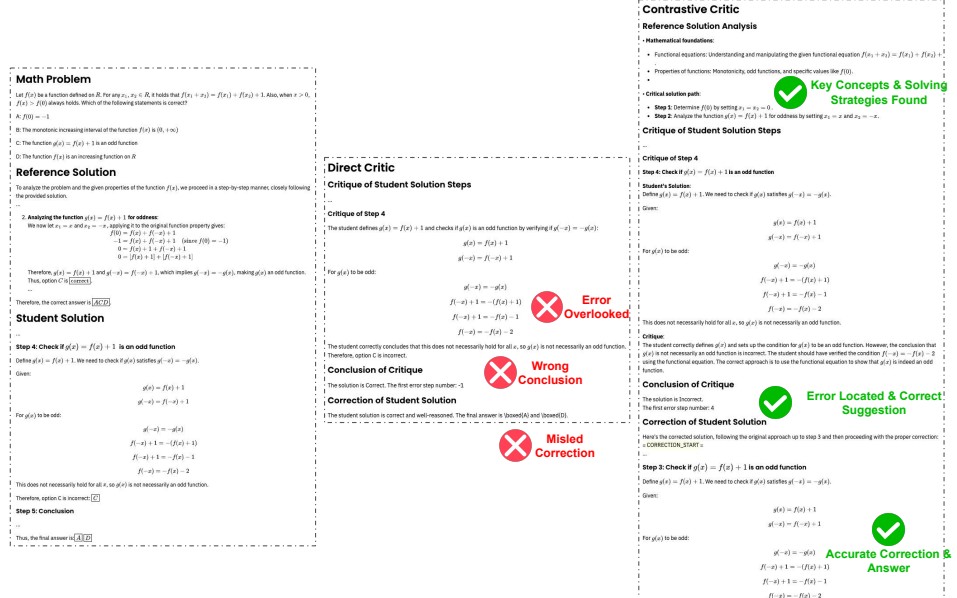

Figure 5: Comparison between Direct Critic and Contrastive Critic. Direct Critic shows blind approval of the student solution, failing to identify any errors and providing misleading approval. In contrast, Contrastive Critic first analyzes the reference solution to understand key mathematical concepts, enabling it to precisely locate the error in the student solution. By developing understanding of the underlying mathematical concepts, Contrastive Critic successfully generate an effective critique that guides the correction process to reach the correct final answer.

# E  Self-Validation Cases

We present two cases demonstrating the effectiveness of our Self-Validation mechanism in filtering critiques based on Self-Critic's correction in Figures 6 and 7.

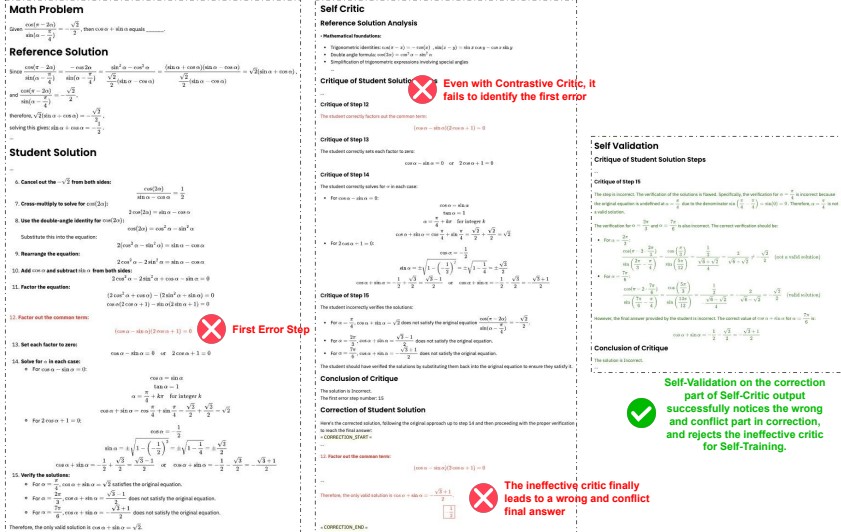

Figure 6: Case1: Self-Validation rejects an ineffective critic: Despite having access to a reference solution and using contrastive learning, the critic fails to identify Step 12 as the first error in solving a trigonometric equation. The subsequent correction leads to a conflicting final answer. The self-validation mechanism successfully detects this inconsistency and rejects this ineffective critique from the training data.

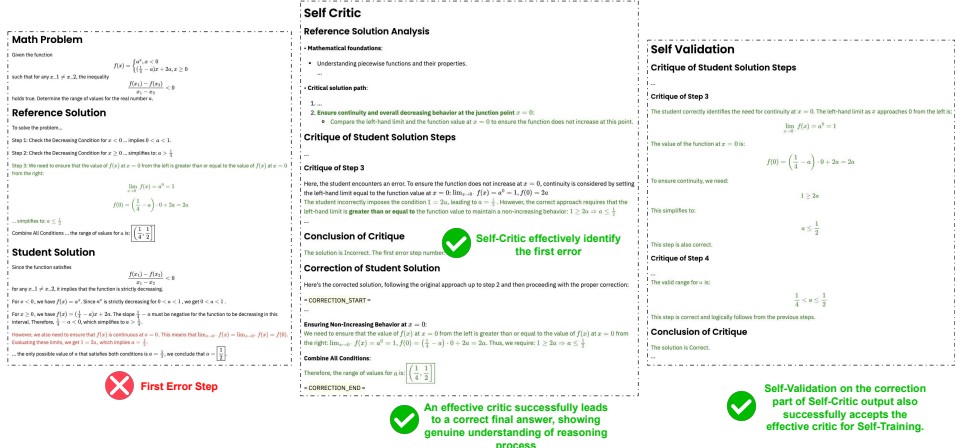

Figure 7: Case2: Self-Validation accepts an effective critic: An example of effective critique that correctly identifies Step 3 as the error point where continuity requirements are mishandled. The correction follows logical mathematical reasoning and arrives at the correct final answer, which is then verified and accepted by the self-validation mechanism for training.

# F  Bug-Injection Case Study

Here we show examples of oversimplified bugs injected by Bug-Injection Critic. These examples illustrate how Bug-Injection Critic tends to generate overly simplistic errors (e.g., misunderstanding basic math properties, variable confusion) rather than more sophisticated mathematical reasoning errors that typically occur in complex problem-solving.

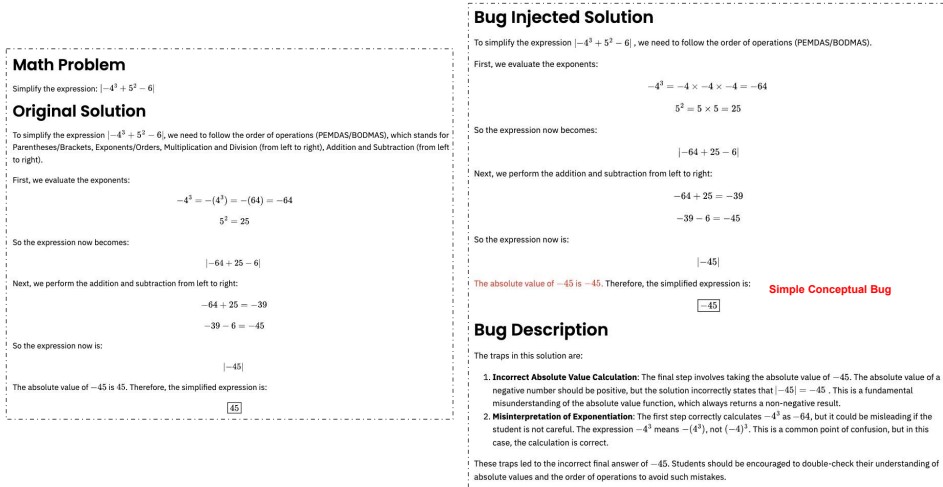

Figure 8: An example of oversimplified bugs injected by Bug-Injection Critic: A conceptual bug involving basic misunderstanding of absolute value property.

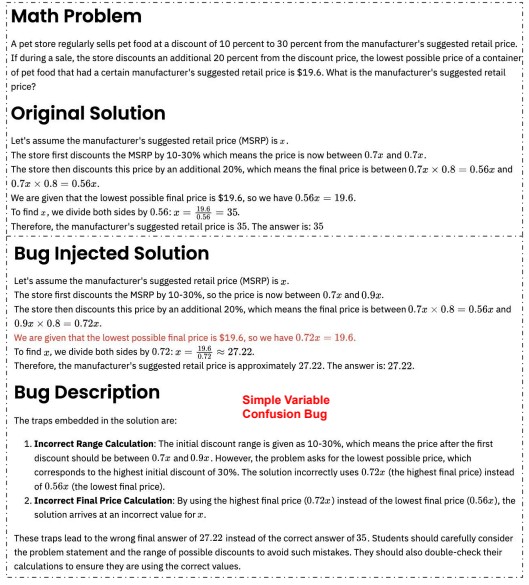

Figure 9: An example of oversimplified bugs injected by Bug-Injection Critic: A variable confusion bug where the wrong price range is used.

# G Adaptations to ProcessBench's Evaluation Protocol

In evaluating models' error identification capabilities, we make two adaptations to ProcessBench's original evaluation protocol. These modifications are designed to ensure that models demonstrate genuine understanding of mathematical errors rather than superficial critique.

## G.1 Requiring Effective Correction

Our first adaptation stems from the core assumption behind critic and correct tasks: a truly effective critique should not only identify errors but also guide their correction towards an correct answer. Through extensive case studies, we found that models can sometimes correctly identify the error step (matching human annotations) without actually understanding the mathematical mistake. As shown in Figures 10 to 12, these cases highlight that merely

matching human-labeled error steps is insufficient for ensuring genuine understanding of mathematical errors.

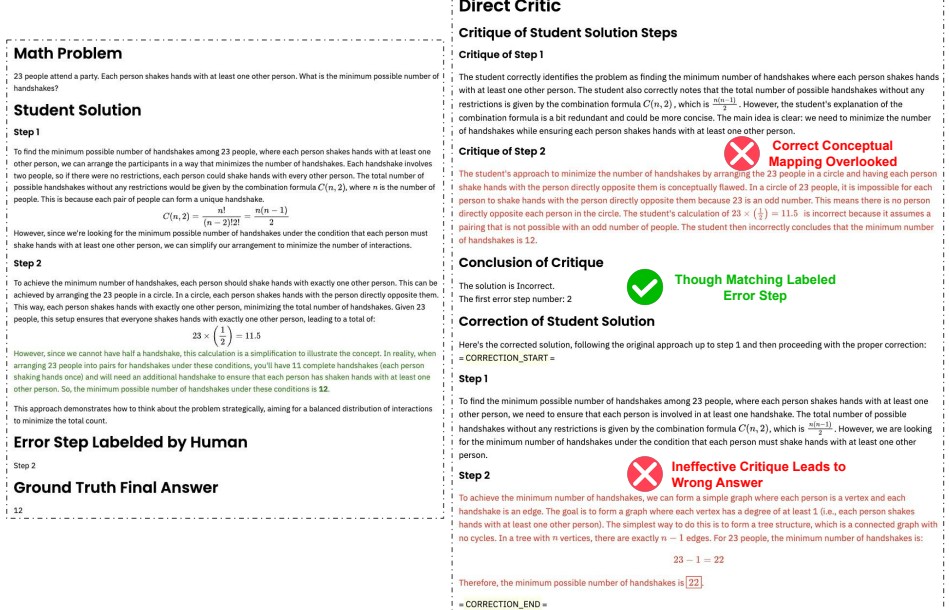

Figure 10: Although the critic correctly identifies Step 2 as the error step (matching human annotation), it fails to understand the underlying mathematical concept of graph theory, leading to an incorrect correction of 22 handshakes instead of the true answer 12.

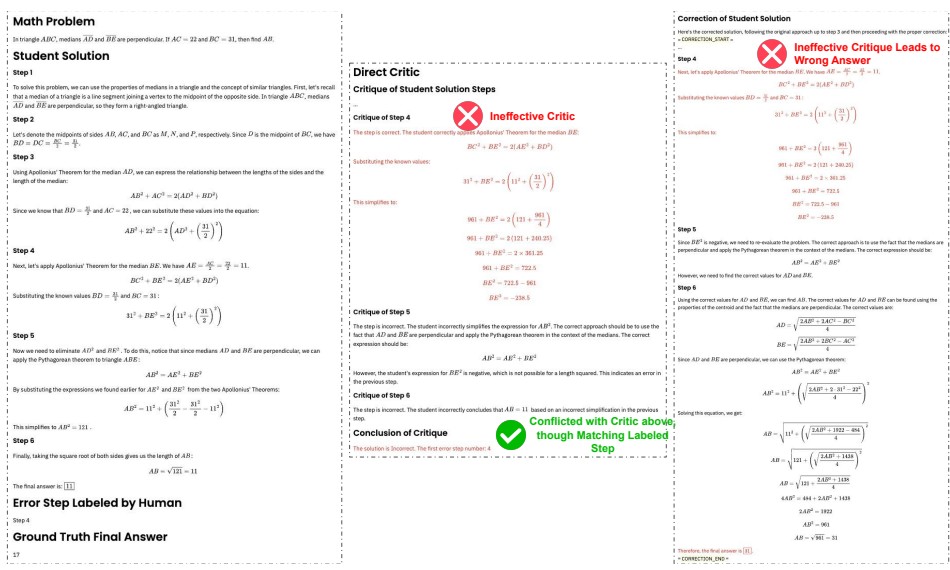

Figure 11: Despite matching the human-labeled error step (Step 4), the critic provides conflicting feedback and fails to recognize the fundamental issue in applying the Pythagorean theorem with perpendicular medians, leading to an incorrect solution.

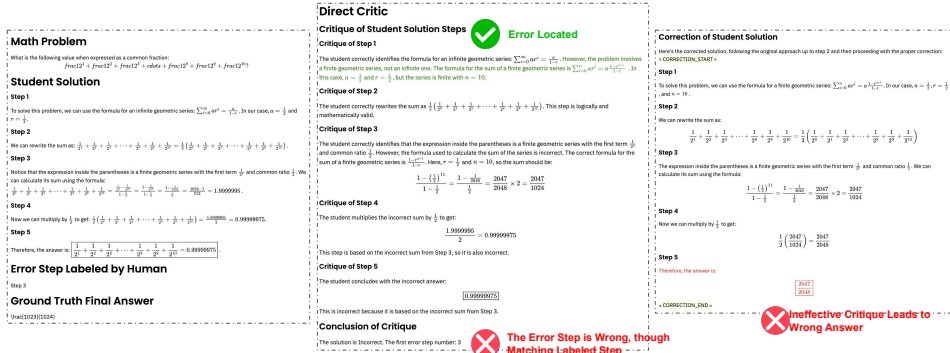

Figure 12: The critic matches Step 3 as problematic but misunderstands the key issue in finite geometric series calculation, resulting in an incorrect final value of 2047/2048.

Therefore, we augment ProcessBench's protocol by requiring that models must not only identify the correct error step but also provide correction that leads to a mathematically valid solution. This stricter requirement helps ensure that models demonstrate genuine understanding of the mathematical concepts and errors involved.

## G.2 Allowing Step-Level Flexibility

Our second adaptation addresses an inherent ambiguity in error identification: in many cases, mathematical errors can reasonably be attributed to multiple consecutive steps. Through our analysis, we found numerous instances where the exact "error step" is debatable, with both the preceding and following steps being valid points of identification. As shown in Figures 13 to 15, these cases illustrate how mathematical errors often span multiple steps, making strict step-level matching overly rigid for meaningful evaluation..

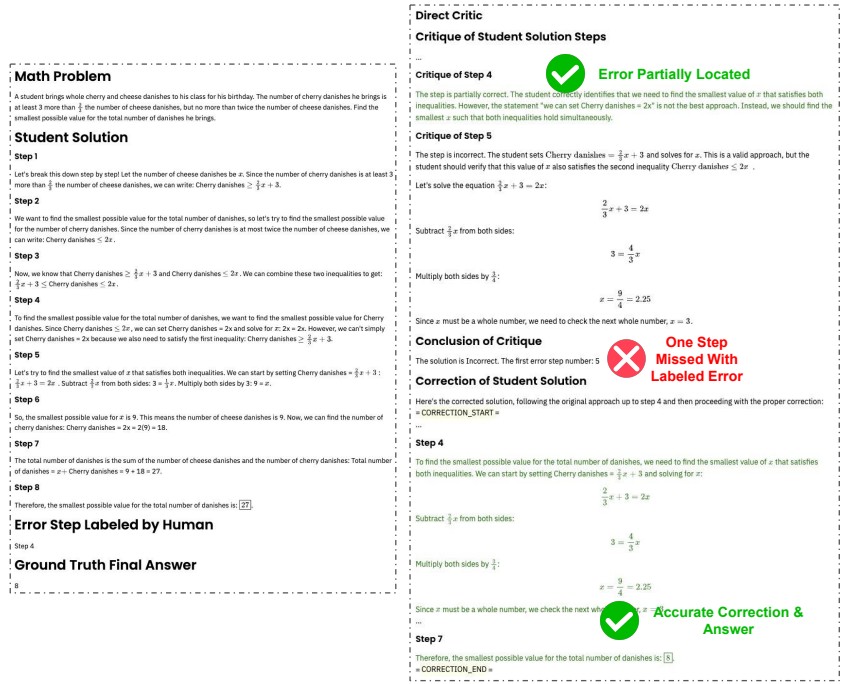

Figure 13: In this cherry-and-cheese danishes problem, while the human annotator labels Step 4 as the error, the true conceptual error begins in Step 5 where the student miscalculates the solution. The model still achieves correct final answer despite identifying a different step.

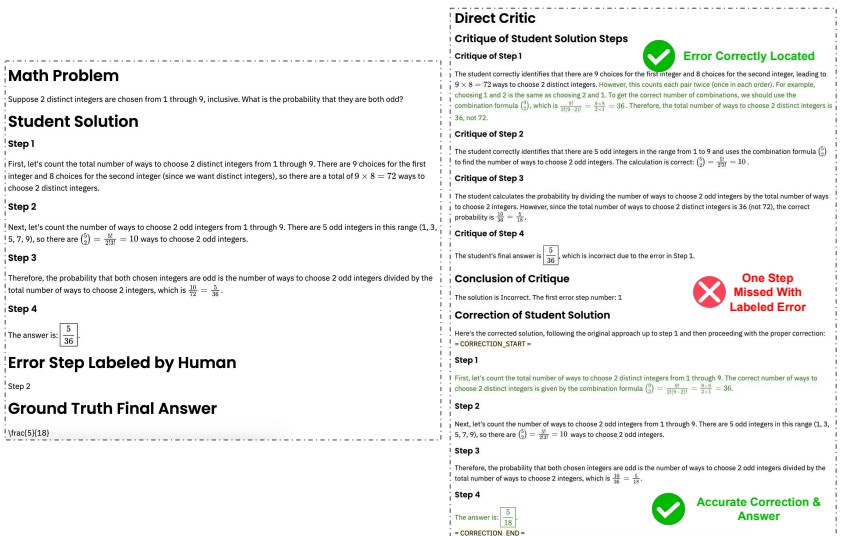

Figure 14: In this probability problem, while the annotator marks Step 2 as the error, the fundamental misconception in Step 1 (overcounting combinations) directly leads to the final incorrect probability.

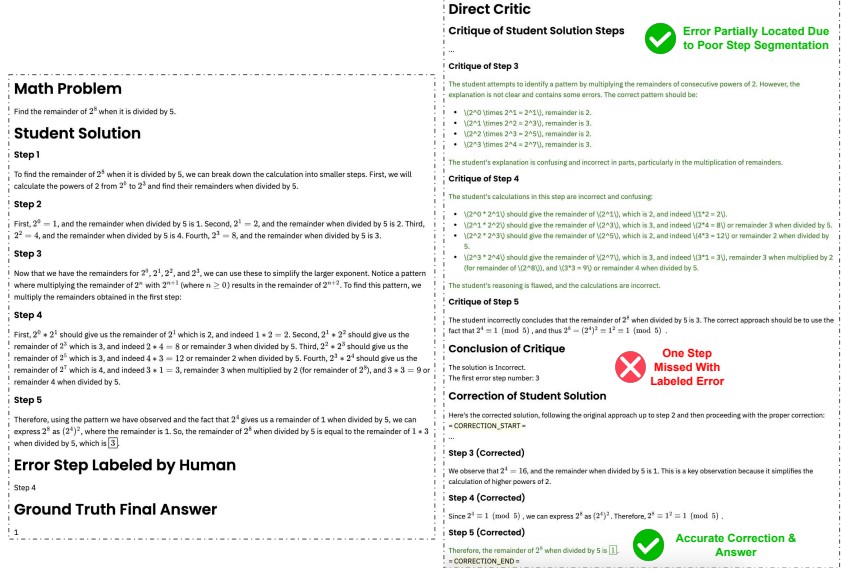

Figure 15: In this remainder calculation problem, the error could be attributed to either Step 3 (pattern identification) or Step 4 (pattern application), as they form a continuous chain of incorrect reasoning.

To account for this ambiguity, we introduce a ±1 step tolerance in matching model predictions with human annotations. This modification better reflects the reality of mathematical error analysis while still maintaining rigor in evaluation.

These adaptations result in a more meaningful evaluation protocol that better captures models' true understanding of mathematical errors and their ability to guide effective corrections.

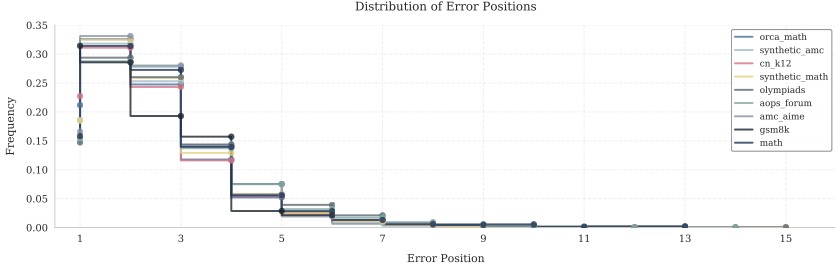

Figure 16: Distribution of first error positions identified by our self-critic across different mathematical domains.

## H  Distribution of First Error Step identified by Self-Critic

## I  Classify Solutions into Correct and Incorrect

Again we use Qwen2.5-72B-Instruct itself to classify solutions into correct and incorrect ones. We present the system prompt in the following Figure 17:

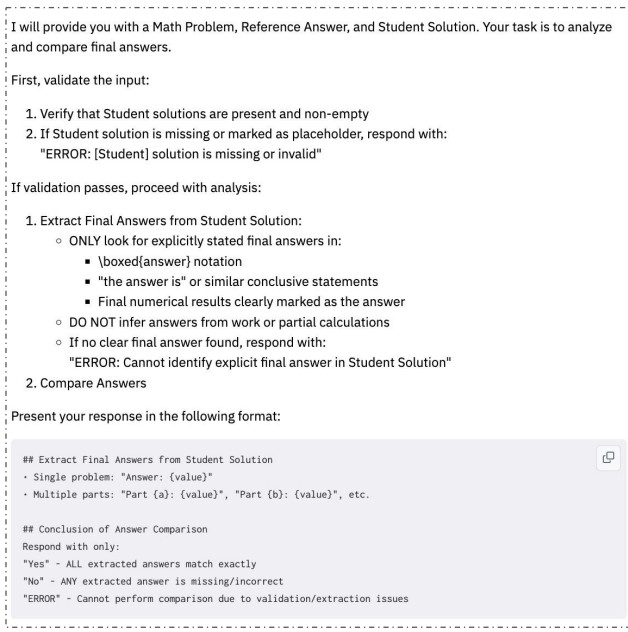

Figure 17: System Prompt to classify solutions into correct and incorrect ones.

## J  Self-Training Details

Here we present the detailed configuration for self-training of Qwen2.5-72B-Instruct. We utilize open-instruct (Wang et al., 2023) for our continued supervised fine-tuning implementation. The training was conducted on 4 servers, each equipped with 8 NVIDIA A100 GPUs (32 GPUs in total), with a total training time of several hours[2].

The key hyper-parameters for training are as follows:

- Batch size: 256

---

[2]The exact training time may vary depending on the specific hardware configuration and system load.

- Learning rate: 5e-6
- Number of training epochs: 1
- Warmup ratio: 0.03
- Model parallel size: 8
- Total GPUs: 32 (4 servers × 8 A100 GPUs)

For reproducibility, we use gradient checkpointing and mixed-precision training (FP16) to optimize memory usage. The training was performed using DeepSpeed ZeRO-3 for efficient distributed training.

