# OpenReview forum: "Self-Evolving Critique Abilities in Large Language Models"
_colmweb.org/COLM/2025/Conference — COLM 2025_

### Official Review · Reviewer_DGVo · 2025-05-11

**Rating:** 6
**Confidence:** 4
**Ethics Flag:** 1

**Summary:**

This submission proposes a self-evolving critic framework, which trains LLMs with self-generated data. The framework also employs a contrastive-critical approach and a self-validation scheme to improve the model quality. It applies the framework to Qwen2.5-72B-instruct model, and the resulting LLM shows consistent improvement on multiple tasks.

**Questions To Authors:**

please see above.

**Reasons To Accept:**

1. A novel self-evolving critique framework that does not require external supervision
2. Consistent performance gains over multiple tasks.

**Reasons To Reject:**

1. Weak baselines: The submission only compares SCRIT with Qwen2.5-72B-instruct and o1-mini, while other strong recent methods such as CriticGPT, Reflexion are not compared.
2. The experiments mainly focus on mathematic problems, while other reasoning tasks, e.g, logical reasoning, planning, and coding, are not considered, while may limit the generalisation ability of the proposed framework.

---

> ### Author Response · Authors · 2025-06-03
>
> > Weak baselines: The submission only compares SCRIT with Qwen2.5-72B-instruct and o1-mini, while other strong recent methods such as CriticGPT, Reflexion are not compared.
>
> Thanks for raising this important question about our baseline selection. We provide a detailed response below:
>
> **Why We Selected Qwen2.5-72B-Instruct and o1-mini as Primary Baselines**:
> - We use Qwen2.5-72B-Instruct as our most critical baseline because it is the base model that our SCRIT framework aims to improve through self-evolution. This direct comparison demonstrates the effectiveness of our self-evolving approach on the target model.
> - o1-mini was selected following ProcessBench [1], as it represents the state-of-the-art accessible model for critique tasks at the time of our submission. This comparison positions our work against the current best-performing model.
>
> **Regarding CriticGPT and Reflexion Comparisons**:
> - **CriticGPT [2]**: We were unable to include CriticGPT in our comparison because it lacks both public API access and open-source model availability, making direct comparison infeasible.
> - **Reflexion [3]**: Due to time constraints, we were unable to implement this approach in our framework during the rebuttal. However, we provide a proxy implementation where the model iteratively critiques over multiple rounds, similar to the idea of Reflexion. We would like to add these results in a future revision.
>
> | Method                                                       | Average Performance |
> | ------------------------------------------------------------ | ------------------- |
> | Qwen2.5-72B-Instruct                                         | 57.7%               |
> | Qwen2.5-72B-Instruct + Iterative Critique (proxy of Reflextion) | 57.3%               |
> | **Qwen2.5-72B-Instruct + SCRIT**                             | **62.1%**           |
>
> We observe that iterative crituqe shows minimal improvement. This finding is consistent with Google DeepMind's ICLR2024 work[4], which demonstrated that self-reflection approaches can be ineffective or even counterproductive for complex reasoning tasks.
>
> We fully acknowledge the importance of comprehensive baseline comparisons and will incorporate these findings into our revised manuscript, including proper citations and a dedicated section comparing our approach with these important prior works.
>
> [1] ProcessBench: Identifying Process Errors in Mathematical Reasoning
> [2] LLM Critics Help Catch LLM Bugs
> [3] Reflexion: Language Agents with Verbal Reinforcement Learning
> [4] Large Language Models Cannot Self-Correct Reasoning Yet

---

> ### Author Response · Authors · 2025-06-03
>
> > The experiments mainly focus on mathematic problems, while other reasoning tasks, e.g, logical reasoning, planning, and coding, are not considered, while may limit the generalisation ability of the proposed framework.
>
> Thanks for your thorough review and helpful suggestions!
>
> **Regarding cross-domain generalization:** Our evaluation already demonstrates the framework's effectiveness beyond mathematical reasoning. As shown in Tables 1 and 2, we evaluated SCRIT across several non-mathematical reasoning benchmarks, including ARC-C (general scientific reasoning tasks including logical reasoning and common sense reasoning), GPQA (graduate-level physics, chemistry, and biology domains), and MMLU-STEM (multidisciplinary STEM knowledge tests). These results indicate that our model, self-evolved on the mathematical domain, generalizes well to these diverse non-mathematical domains, maintaining consistent effectiveness across different reasoning tasks.
>
> **Additional Results on Scientific Reasoning Experiments:** To further address this concern, during the rebuttal period, we conducted additional experiments extending SCRIT to **diverse scientific reasoning domains**. We systematically collected data from AM-Thinking-v1 [1] spanning multiple scientific disciplines including physics, chemistry, and natural sciences, as well as logical reasoning tasks.
>
> Following the exact same methodology as our mathematical reasoning approach, we synthesized 10K critique training examples and applied identical training configurations. Results are reported below:
>
> | Model                     | CC-Acc | EI-F1 |
> | ------------------------- | ------ | ----- |
> | Qwen2.5-72B-Instruct      | 53.0   | 37.8  |
> | + SCRIT (Math Data)       | 58.3   | 45.1  |
> | + SCRIT (Scientific Data) | 59.6   | 42.5  |
>
> The scientific reasoning data achieves **comparable or even superior performance**, demonstrating that our approach generalizes effectively to non-mathematical domains. We also provide a detailed breakdown by domain groups below:
>
> | **Model**                                         | **Math Reasoning** | **Scientific Reasoning** | **Overall Avg** |
> | ------------------------------------------------- | ------------------ | ------------------------ | --------------- |
> | **Critic on deliberately incorrect solutions**    |                    |                          |                 |
> | Qwen2.5-72B-Instruct                              | 38.8               | 41.4                     | 39.7            |
> | + SCRIT (Math Data)                               | 47.5               | 54.2                     | 50.0            |
> | + SCRIT (Scientific Data)                         | 44.0               | **62.8**                 | **51.0**        |
> | **Critic on balanced solutions**                  |                    |                          |                 |
> | Qwen2.5-72B-Instruct                              | 60.2               | 52.8                     | 57.7            |
> | + SCRIT (Math Data)                               | 64.5               | 61.4                     | 62.1            |
> | + SCRIT (Scientific Data)                         | 61.8               | **67.4**                 | **63.9**        |
> | **Critic on Qwen2.5-72B-Instruct's own solution** |                    |                          |                 |
> | Qwen2.5-72B-Instruct                              | 61.6               | 61.9                     | 61.7            |
> | + SCRIT (Math Data)                               | 63.7               | 64.7                     | 62.9            |
> | + SCRIT (Scientific Data)                         | 60.3               | **69.7**                 | **63.9**        |
>
> *Math Reasoning includes: GSM8K, MATH, CollegeMath, MinervaMath, OlympiadBench*
> *Scientific Reasoning includes: ARC-C, GPQA, MMLU-STEM*
>
> Notably, SCRIT trained on scientific reasoning data achieves **significantly larger improvements on scientific benchmarks** while maintaining competitive performance on mathematical tasks, confirming both within-domain effectiveness and strong cross-domain transfer capabilities.
>
> [1] AM-Thinking-v1: Advancing the Frontier of Reasoning at 32B Scale

---

> > ### Comment · Reviewer_DGVo · 2025-06-03
> >
> > Thanks to the authors for clarification.
> > Given the clarification and new results, I am happy to raise the score.

---

> > > ### Author Response · Authors · 2025-06-04
> > >
> > > Thank you very much for taking the time to read our rebuttal! We sincerely appreciate your thoughtful and constructive feedback, which has significantly helped strengthen our work. Your adjustment to the score is greatly appreciated!

---

### Official Review · Reviewer_jYD3 · 2025-05-12

**Rating:** 6
**Confidence:** 3
**Ethics Flag:** 1

**Summary:**

The study introduces SCRIT, a self-supervised framework that enhances LLMs' abilities to critique and correct solutions using only self-generated data. Experiments show that SCRIT improves error detection and correction accuracy without relying on human or stronger model supervision.

**Reasons To Accept:**

1. This paper is well-motivated and easy to understand.
2. The proposed SCRIT introduces a self-evolving critique mechanism that eliminates the need for human or stronger model supervision, addressing an important challenge in scalable oversight.
3. SCRIT achieves consistent and significant improvements across multiple math benchmarks.

**Reasons To Reject:**

1. Reliance on Reference Solutions:
While SCRIT avoids the need for human supervision, it still relies on access to correct reference solutions. This may not be practical in many real-world scenarios where such reference solutions are unavailable or difficult to obtain.

2. Limited Domain Evaluation:
The evaluation is restricted to mathematical reasoning tasks, where reference solutions are well-defined and readily accessible. It would be beneficial to validate the approach on tasks from other domains, particularly those where reference answers are ambiguous or not easily determined.

---

> ### Author Response · Authors · 2025-06-03
>
> > Reliance on Reference Solutions: While SCRIT avoids the need for human supervision, it still relies on access to correct reference solutions. This may not be practical in many real-world scenarios where such reference solutions are unavailable or difficult to obtain.
>
> Thanks for your feedback and insightful comments!
>
> We'd like to address two key points that may clarify the scope and applicability of our approach.
>
> **Regarding the availability of reference solutions:** We respectfully suggest that reference solutions are more widely available than initially apparent. Many professional domains beyond mathematics possess well-defined correct answers and standardized solution paths. For example, chemistry, biology, and engineering fields commonly employ standardized tests and certifications with established correct solutions. Moreover, we view the reliance on reference solutions not as a limitation, but as a necessary component for effective critique generation. Our empirical results demonstrate that without reference solutions, Direct Critic plateaus at 55.1% CC-Acc and 38.7% EI-F1 with limited scaling potential (Figure 3).
>
> **Regarding cross-domain generalization:** Our evaluation already demonstrates the framework's effectiveness beyond mathematical reasoning. As shown in Tables 1 and 2, we evaluated SCRIT across several non-mathematical reasoning benchmarks, including ARC-C (general scientific reasoning tasks including logical reasoning and common sense reasoning), GPQA (graduate-level physics, chemistry, and biology domains), and MMLU-STEM (multidisciplinary STEM knowledge tests). These results indicate that our approach successfully generalizes across diverse domains while maintaining effectiveness.
>
> **Additional Results on Scientific Reasoning Experiments:** To further address this concern, during the rebuttal period, we conducted additional experiments extending SCRIT to **diverse scientific reasoning domains**. We systematically collected data from AM-Thinking-v1 [1] spanning multiple scientific disciplines including physics, chemistry, and natural sciences, as well as logical reasoning tasks.
>
> Following the exact same methodology as our mathematical reasoning approach, we synthesized 10K critique training examples and applied identical training configurations. Results are reported below:
>
> | Model                     | CC-Acc | EI-F1 |
> | ------------------------- | ------ | ----- |
> | Qwen2.5-72B-Instruct      | 53.0   | 37.8  |
> | + SCRIT (Math Data)       | 58.3   | 45.1  |
> | + SCRIT (Scientific Data) | 59.6   | 42.5  |
>
> The scientific reasoning data achieves **comparable or even superior performance**, demonstrating that our approach generalizes effectively to non-mathematical domains. We also provide a detailed breakdown by domain groups below:
>
> | **Model**                                         | **Math Reasoning** | **Scientific Reasoning** | **Overall Avg** |
> | ------------------------------------------------- | ------------------ | ------------------------ | --------------- |
> | **Critic on deliberately incorrect solutions**    |                    |                          |                 |
> | Qwen2.5-72B-Instruct                              | 38.8               | 41.4                     | 39.7            |
> | + SCRIT (Math Data)                               | 47.5               | 54.2                     | 50.0            |
> | + SCRIT (Scientific Data)                         | 44.0               | **62.8**                 | **51.0**        |
> | **Critic on balanced solutions**                  |                    |                          |                 |
> | Qwen2.5-72B-Instruct                              | 60.2               | 52.8                     | 57.7            |
> | + SCRIT (Math Data)                               | 64.5               | 61.4                     | 62.1            |
> | + SCRIT (Scientific Data)                         | 61.8               | **67.4**                 | **63.9**        |
> | **Critic on Qwen2.5-72B-Instruct's own solution** |                    |                          |                 |
> | Qwen2.5-72B-Instruct                              | 61.6               | 61.9                     | 61.7            |
> | + SCRIT (Math Data)                               | 63.7               | 64.7                     | 62.9            |
> | + SCRIT (Scientific Data)                         | 60.3               | **69.7**                 | **63.9**        |
>
> *Math Reasoning includes: GSM8K, MATH, CollegeMath, MinervaMath, OlympiadBench*
> *Scientific Reasoning includes: ARC-C, GPQA, MMLU-STEM*
>
> Notably, SCRIT trained on scientific reasoning data achieves **significantly larger improvements on scientific benchmarks** while maintaining competitive performance on mathematical tasks, confirming both within-domain effectiveness and strong cross-domain transfer capabilities.
>
> [1] AM-Thinking-v1: Advancing the Frontier of Reasoning at 32B Scale

---

> > ### Comment · Reviewer_jYD3 · 2025-06-06
> >
> > Thank you for your clarification and for conducting the new experiments. I intend to maintain my current score, which is in favor of the acceptance of this paper.

---

> > > ### Author Response · Authors · 2025-06-08
> > >
> > > Thank you for taking the time to read our responses and thanks again for your review! Should you have any further comments, please feel free to let us know.

---

> ### Author Response · Authors · 2025-06-03
>
> > Limited Domain Evaluation: The evaluation is restricted to mathematical reasoning tasks, where reference solutions are well-defined and readily accessible. It would be beneficial to validate the approach on tasks from other domains, particularly those where reference answers are ambiguous or not easily determined.
>
> Thanks for your constructive feedback and important observations! We guess there may be some misunderstanding of our framework, and we would like to first clarify a key distinction in how reference solutions are used in our framework. Then, we will address your concerns about generality.
>
> **Clarification: Reference Solutions Only Used During Data Synthesis, Not in Evaluation**.  There is an important distinction between **data synthesis**, **training**, and **testing** phases in our approach:
>
> - **Data Synthesis Phase:** During data synthesis (Section 2.3), we use reference solutions to improve the quality of generated critiques. Given a problem `p` and student solution `s_student`, we introduce reference solution `s_reference` to help the model generate higher-quality critique `c` that includes: reference analysis + step-wise critique + conclusion + correction.
> - **Training Phase:** During training, our **input contains only the problem `p` and student solution `s_student`**. The **target output excludes the reference analysis** and consists of: step-wise critique + conclusion + correction. Crucially, **no reference solution is provided as input during training**.
> - **Testing Phase:**  During evaluation, our model takes **only the problem `p` and student solution `s_student` as input**, without any reference solution dependency. This makes our approach practical for real-world scenarios where reference solutions may not be available during inference.
>
> Thus, **reference solutions are only needed during the initial data synthesis phase to bootstrap high-quality critique examples**. Once trained, the model can critique solutions in new domains without requiring reference solutions, making it practical for real-world applications where such references may be unavailable.
>
> **Cross-Domain Generalization Evidence**:  As demonstrated in our response to the previous concern, we have conducted extensive experiments showing generalization to scientific reasoning domains:
>
> - **Training data**: We successfully synthesized critique data from scientific reasoning domains using the same methodology
> - **Evaluation**: SCRIT shows strong performance on non-mathematical benchmarks (ARC-C, GPQA, MMLU-STEM)
> - **Cross-domain transfer**: Models trained on mathematical data perform well on scientific tasks, and vice versa

---

### Official Review · Reviewer_whUs · 2025-05-12

**Rating:** 7
**Confidence:** 4
**Ethics Flag:** 1

**Summary:**

This paper presents SCRIT, a framework to improve the critique abilities of LLMs without relying on supervision from humans or stronger models. The authors' proposed method includes 1) a contrastive critic approach, where the model critiques a given solution by comparing it to a correct reference solution, and 2) a self-validation stage, which ensures that only critiques which lead to verifiably correct corrections are used for training. The authors evaluate their approach on mathematical reasoning, showing improvements on critic and correction tasks as well as error identification tasks, compared to the base model (Qwen2.5-72B-Instruct).

**Questions To Authors:**

Questions:
- In Tables 1 and 2, what do the bold numbers indicate? In some columns it is the highest number, whereas in others it is not.
- In Table 1, what is your hypothesis for SCRIT giving more improvements on some datasets (e.g., MMLU) vs others?
- In Table 1, under "critic on balanced solutions", SCRIT seems to hurt performance on 3 out of 8 datasets compared to the base model. Why is that?
- Have you done any qualitative analysis of the generated critiques? Are there any systematic failure patterns observed in SCRIT generated critiques?
- How does the choice of the base-model affect the outcome? Can SCRIT improve smaller models?

Presentation:
- The problem formulation is for mathematical problems (section 2), however, in the introduction, it is suggested that the approach is general but the authors decided to apply it to math as a testbed.

**Reasons To Accept:**

- The topic is relevant and the paper is well-written and motivated. Given the focus on mathematical reasoning, the evaluations are sufficient to show improvements compared to the base model, and illustrate the gap to SOTA performance.
- The proposed method is straightforward and generates quality critique data without relying on human annotators or stronger models, making it more scalable and cost-effective.

**Reasons To Reject:**

- The approach is math focused and depends on the availability of reference correct solutions, which limits generalization. It is not clear if the approach can be extended to other domains and more open-ended problems.
- Some prior work in this domain are not referenced in the paper. For example:
   - "Shepherd: A Critic for Language Model Generation" (Wang et al., 2023)
   - "CriticEval: Evaluating Large Language Model as Critic" (Lan et al. 2024)

---

> ### Author Response · Authors · 2025-06-03
>
> > The approach is math focused and depends on the availability of reference correct solutions, which limits generalization. It is not clear if the approach can be extended to other domains and more open-ended problems.
>
> Thank you for your thoughtful feedback and important questions about our framework's generalizability. We'd like to address two key points that may clarify the scope and applicability of our approach.
>
> **Regarding the availability of reference solutions:** We respectfully suggest that reference solutions are more widely available than initially apparent. Many professional domains beyond mathematics possess well-defined correct answers and standardized solution paths. For example, chemistry, biology, and engineering fields commonly employ standardized tests and certifications with established correct solutions. Moreover, we view the reliance on reference solutions not as a limitation, but as a necessary component for effective critique generation. Our empirical results demonstrate that without reference solutions, Direct Critic plateaus at 55.1% CC-Acc and 38.7% EI-F1 with limited scaling potential (Figure 3).
>
> **Regarding cross-domain generalization:** Our evaluation already demonstrates the framework's effectiveness beyond mathematical reasoning. As shown in Tables 1 and 2, we evaluated SCRIT across several non-mathematical reasoning benchmarks, including ARC-C (general scientific reasoning tasks including logical reasoning and common sense reasoning), GPQA (graduate-level physics, chemistry, and biology domains), and MMLU-STEM (multidisciplinary STEM knowledge tests). These results indicate that our approach successfully generalizes across diverse domains while maintaining effectiveness.
>
> **Additional Results on Scientific Reasoning Experiments:** To further address this concern, during the rebuttal period, we conducted additional experiments extending SCRIT to **diverse scientific reasoning domains**. We systematically collected data from AM-Thinking-v1 [1] spanning multiple scientific disciplines including physics, chemistry, and natural sciences, as well as logical reasoning tasks.
>
> Following the exact same methodology as our mathematical reasoning approach, we synthesized 10K critique training examples and applied identical training configurations. Results are reported below:
>
> | Model                     | CC-Acc | EI-F1 |
> | ------------------------- | ------ | ----- |
> | Qwen2.5-72B-Instruct      | 53.0   | 37.8  |
> | + SCRIT (Math Data)       | 58.3   | 45.1  |
> | + SCRIT (Scientific Data) | 59.6   | 42.5  |
>
> The scientific reasoning data achieves **comparable or even superior performance**, demonstrating that our approach generalizes effectively to non-mathematical domains. We also provide a detailed breakdown by domain groups below:
>
> | **Model**                                         | **Math Reasoning** | **Scientific Reasoning** | **Overall Avg** |
> | ------------------------------------------------- | ------------------ | ------------------------ | --------------- |
> | **Critic on deliberately incorrect solutions**    |                    |                          |                 |
> | Qwen2.5-72B-Instruct                              | 38.8               | 41.4                     | 39.7            |
> | + SCRIT (Math Data)                               | 47.5               | 54.2                     | 50.0            |
> | + SCRIT (Scientific Data)                         | 44.0               | **62.8**                 | **51.0**        |
> | **Critic on balanced solutions**                  |                    |                          |                 |
> | Qwen2.5-72B-Instruct                              | 60.2               | 52.8                     | 57.7            |
> | + SCRIT (Math Data)                               | 64.5               | 61.4                     | 62.1            |
> | + SCRIT (Scientific Data)                         | 61.8               | **67.4**                 | **63.9**        |
> | **Critic on Qwen2.5-72B-Instruct's own solution** |                    |                          |                 |
> | Qwen2.5-72B-Instruct                              | 61.6               | 61.9                     | 61.7            |
> | + SCRIT (Math Data)                               | 63.7               | 64.7                     | 62.9            |
> | + SCRIT (Scientific Data)                         | 60.3               | **69.7**                 | **63.9**        |
>
> *Math Reasoning includes: GSM8K, MATH, CollegeMath, MinervaMath, OlympiadBench*
> *Scientific Reasoning includes: ARC-C, GPQA, MMLU-STEM*
>
> Notably, SCRIT trained on scientific reasoning data achieves **significantly larger improvements on scientific benchmarks** while maintaining competitive performance on mathematical tasks, confirming both within-domain effectiveness and strong cross-domain transfer capabilities.
>
> [1] AM-Thinking-v1: Advancing the Frontier of Reasoning at 32B Scale

---

> ### Author Response · Authors · 2025-06-03
>
> > In Table 1, what is your hypothesis for SCRIT giving more improvements on some datasets (e.g., MMLU) vs others?
>
> Thanks for your thorough review and interesting questions! We hypothesize that SCRIT achieves larger improvements on MMLU-STEM compared to other datasets due to two key factors:
>
> -  **Constrained Answer Space in Multiple-Choice Format**: MMLU presents problems in a multiple-choice format with four options, which changes the critique task dynamics. When critiquing a solution, the model can systematically evaluate each option against the reasoning steps and leverage process of elimination to identify errors more effectively. In contrast, open-ended datasets like MATH require generating corrections from an unbounded solution space.
> - **Problem Difficulty and Complexity**  MMLU-STEM generally presents problems at an undergraduate level, which differs from graduate-level GPQA problems or competition-level MATH and Olympiad problems. The relatively moderate difficulty allows SCRIT to better identify the root causes of errors and generate more accurate corrections due to clearer conceptual boundaries.

---

> ### Author Response · Authors · 2025-06-03
>
> > In Table 1, under "critic on balanced solutions", SCRIT seems to hurt performance on 3 out of 8 datasets compared to the base model. Why is that?
>
> > Have you done any qualitative analysis of the generated critiques? Are there any systematic failure patterns observed in SCRIT generated critiques?
>
> We appreciate your insightful feedback and constructive comments. To address both questions comprehensively, we conducted a systematic qualitative analysis.
>
> **Clarification on Performance Changes** First, we note that the performance differences are quite comparable: CollegeMath (50.9% vs 50.5%), GPQA (31.1% vs 29.5%), and MinervaMath (47.1% vs 45.6%). To better understand these small variations, we conducted a detailed qualitative analysis.
>
> **Qualitative Analysis of Failure Patterns**: To understand these performance drops, we conducted a systematic case study by randomly sampling 8 QA pairs from each test set and analyzing the critiques generated by both the baseline Qwen2.5-72B-Instruct and SCRIT models. We defined five failure dimensions:
>
> 1. **Failed Error Detection**: Did the model fail to identify where errors are in the student solution?
> 2. **Failed Root Cause Analysis**: Did the model identify error locations but fail to explain why they are errors?
> 3. **Failed Correction Guidance**: Did the model identify errors and causes but fail to provide proper correction suggestions?
> 4. **Conclusion Inconsistency**: Is there inconsistency between step-by-step critique and final conclusion?
> 5. **Correction Misalignment**: Does the corrected solution deviate from the correction suggestions?
>
> Results are provided below (64 total samples: 8 samples × 8 datasets, showing percentage of total samples with each error type):
>
> | Error Type                     | GSM8K | MATH | CollegeMath | MinervaMath | OlympiadBench | ARC-C | GPQA | MMLU-STEM | Total |
> | ------------------------------ | ----- | ---- | ----------- | ----------- | ------------- | ----- | ---- | --------- | ----- |
> | **Qwen2.5-72B-Instruct**       |       |      |             |             |               |       |      |           |       |
> | Failed Error Detection         | 0%    | 3.1% | 1.6%        | 9.4%        | 9.4%          | 1.6%  | 3.1% | 1.6%      | 29.7% |
> | Failed Root Cause Analysis     | 0%    | 0%   | 1.6%        | 1.6%        | 3.1%          | 1.6%  | 3.1% | 1.6%      | 12.5% |
> | Failed Correction Guidance     | 0%    | 0%   | 3.1%        | 1.6%        | 6.3%          | 1.6%  | 6.3% | 3.1%      | 21.9% |
> | Conclusion Inconsistency       | 0%    | 0%   | 1.6%        | 1.6%        | 3.1%          | 0%    | 1.6% | 0%        | 7.8%  |
> | Correction Misalignment        | 0%    | 0%   | 3.1%        | 3.1%        | 6.3%          | 0%    | 6.3% | 1.6%      | 20.3% |
> | **Qwen2.5-72B-Instruct+SCRIT** |       |      |             |             |               |       |      |           |       |
> | Failed Error Detection         | 0%    | 1.6% | 1.6%        | 4.7%        | 6.3%          | 1.6%  | 3.1% | 1.6%      | 20.3% |
> | Failed Root Cause Analysis     | 0%    | 0%   | 1.6%        | 3.1%        | 4.7%          | 0%    | 3.1% | 0%        | 12.5% |
> | Failed Correction Guidance     | 0%    | 0%   | 3.1%        | 4.7%        | 6.3%          | 1.6%  | 6.3% | 0%        | 21.9% |
> | Conclusion Inconsistency       | 0%    | 0%   | 1.6%        | 3.1%        | 1.6%          | 0%    | 1.6% | 1.6%      | 9.4%  |
> | Correction Misalignment        | 0%    | 0%   | 3.1%        | 1.6%        | 6.3%          | 0%    | 3.1% | 0%        | 14.1% |
>
> **Key findings and Systematic Failure Patterns**: The analysis reveals both improvements and persistent challenges:
>
> **SCRIT's Improvements:**
> - **Better Error Detection**: Reduces failed error detection from 29.7% to 20.3%
> - **Improved Correction Alignment**: Reduces correction misalignment from 20.3% to 14.1%
>
> **Systematic Failure Patterns:**
> 1. **Complex Reasoning Limitations**: For highly complex datasets like MinervaMath and GPQA, SCRIT shows increases in Failed Root Cause Analysis and Conclusion Inconsistency, indicating struggles with graduate-level mathematical reasoning.
>
> 2. **Correction Guidance Gaps**: Failed correction guidance remains unchanged (21.9%), suggesting that even when SCRIT identifies errors correctly, translating understanding into actionable correction strategies remains challenging.
>
> This explains the slight performance drops on 3 datasets: SCRIT's improvements are not uniform across all problem types. For graduate-level problems requiring deep mathematical reasoning, the model still faces systematic challenges in providing coherent and effective critiques, leading to slightly degraded performance in the balanced solutions scenario.

---

> ### Author Response · Authors · 2025-06-03
>
> > How does the choice of the base-model affect the outcome? Can SCRIT improve smaller models?
>
> Thanks for raising this important question about the generalizability of SCRIT across different model sizes! To address your concern about whether SCRIT can improve smaller models, we conducted additional experiments using **Qwen2.5-32B-Instruct** as the base model. We followed the exact same methodology: using the 32B model for self-evolving data synthesis, applying identical training hyperparameters, and evaluating on the same benchmarks.
>
> **Results on smaller models**:
>
> | Model                      | CC-Acc | EI-F1 |
> | -------------------------- | ------ | ----- |
> | Qwen2.5-72B-Instruct       | 53.0   | 37.8  |
> | Qwen2.5-72B-Instruct+SCRIT | 58.3   | 45.1  |
> | Qwen2.5-32B-Instruct       | 53.9   | 35.8  |
> | Qwen2.5-32B-Instruct+SCRIT | 56.5   | 41.5  |
>
> The results demonstrate that **SCRIT successfully improves smaller models** with meaningful gains:
> - **CC-Acc**: +2.6 points improvement (53.9% → 56.5%)
> - **EI-F1**: +5.7 points improvement (35.8% → 41.5%)
>
> Notably, the 32B model shows **proportionally similar improvements** to the 72B model, indicating that SCRIT's self-evolving mechanism is robust across different model scales. We will include these results in the updated manuscript to provide a more comprehensive evaluation of our approach.

---

> ### Author Response · Authors · 2025-06-03
>
> > Some prior work in this domain are not referenced in the paper.
>
> We appreciate you pointing out these important missing references. We will include comprehensive citations and discussions of the following works in our updated manuscript:
> - "Shepherd: A Critic for Language Model Generation" (Wang et al., 2023)
> - "CriticEval: Evaluating Large Language Model as Critic" (Lan et al. 2024)
>
> These works represent important contributions to the critique evaluation domain, and we will properly position our work in relation to these efforts in the related work section.
>
> > In Tables 1 and 2, what do the bold numbers indicate? In some columns it is the highest number, whereas in others it is not.
>
> Thank you for pointing out this formatting inconsistency. The bold numbers were intended to highlight cases where SCRIT shows improvement over the base model (Qwen2.5-72B-Instruct), rather than indicating the highest values across all models. We acknowledge this may cause confusion for readers and will revise the table formatting in the updated manuscript to make the comparison criteria clear and consistent.
>
> > The problem formulation is for mathematical problems (section 2), however, in the introduction, it is suggested that the approach is general but the authors decided to apply it to math as a testbed.
>
> You raise an important point about the apparent inconsistency between our general claims in the introduction and the mathematical focus in the problem formulation. We will clarify the scope and necessary conditions of our approach more precisely in the revised manuscript. Specifically, we will:
>
> 1. **Better articulate the applicability conditions**: Our method applies to domains with verifiable reference solutions, which extends beyond mathematics to many structured reasoning tasks.
>
> 2. **Include cross-domain experimental results**: We will incorporate our scientific reasoning experiments (as demonstrated in our earlier responses) into the main methodology and experimental sections to provide concrete evidence of generalizability.
>
> 3. **Refine the introduction**: We will be more precise about the scope of our claims while maintaining the broader vision of the approach's potential applications.
>
> These revisions will provide a more accurate and comprehensive presentation of our work's contributions and limitations.
>
> ---
>
> Thank you for your feedback. We sincerely appreciate your time and would be grateful if you could re-evaluate our work based on the above responses. If you have any additional concerns, we would be happy to address them.

---

> ### Author Response · Authors · 2025-06-08
>
> Dear Reviewer,
>
> Thank you for your thoughtful feedback on our paper. We've carefully addressed your concerns in our rebuttal and hope our responses are helpful.
>
> We'd appreciate hearing your thoughts when you have a chance!
>
> Thanks again for your time and consideration!
>
> Best regards,
>
> Authors

---

> > ### Comment · Reviewer_whUs · 2025-06-09
> >
> > Thanks for the detailed response to my questions. The authors have addressed my concerns, as such I'll raise my score.

---

> > > ### Author Response · Authors · 2025-06-09
> > >
> > > Thank you so much for taking the time to read our detailed rebuttal! We sincerely appreciate your thoughtful consideration and your decision to raise the score.
> > >
> > > Your constructive feedback has been invaluable in strengthening our work. We will incorporate all the discussions and additional results from our rebuttal into the revised manuscript.
> > >
> > > Thank you again for your supportive evaluation!

---

### Decision · Program_Chairs · 2025-07-08

**Decision:**

Accept

**Comment:**

This paper presents SCRIT, a framework to improve the critique abilities of LLMs without relying on supervision from humans or stronger models. The authors' proposed method includes 1) a contrastive critic approach, where the model critiques a given solution by comparing it to a correct reference solution, and 2) a self-validation stage, which ensures that only critiques which lead to verifiably correct corrections are used for training. The authors evaluate their approach on mathematical reasoning, showing improvements on critic and correction tasks as well as error identification tasks, compared to the base model (Qwen2.5-72B-Instruct).

Reasons To Accept:
* The topic is relevant and the paper is well-written and motivated. Given the focus on mathematical reasoning, the evaluations are sufficient to show improvements compared to the base model, and illustrate the gap to SOTA performance.
* The proposed method is straightforward and generates quality critique data without relying on human annotators or stronger models, making it more scalable and cost-effective.
* The proposed SCRIT introduces a self-evolving critique mechanism, addressing an important challenge in scalable oversight.
* SCRIT achieves consistent and significant improvements across multiple math benchmarks.

Reasons To Reject:
* Reliance on Reference Solutions: While SCRIT avoids the need for human supervision, it still relies on access to correct reference solutions. This may not be practical in many real-world scenarios where such reference solutions are unavailable or difficult to obtain.
* Limited Domain Evaluation: The evaluation is restricted to mathematical reasoning tasks, where reference solutions are well-defined and readily accessible. It would be beneficial to validate the approach on tasks from other domains, particularly those where reference answers are ambiguous or not easily determined.